# Hard input, soft management and student engagement: How institutional actions promote innovation ability and academic performance among top innovative talent

Tianbao Zhang, Weihai Huang\*, Zhimin Liu\*

College of Public Administration, Nanjing Agricultural University, Nanjing, Jiangsu Province, China

\* huangweihai@njau.edu.cn (WH); liuzhimin@njau.edu.cn (ZL)

## Abstract

The training and cultivation of innovative talent, especially top-quality talent, cannot be explained fully by reference to a simple linear input–output model. Research that can open the "black box" that characterizes this process has important theoretical and practical significance with respect to efforts to promote the development of high-quality education. By reference to a sample consisting of 1,500 graduating doctoral students who participated in a national stratified sampling survey in China, we develop a structural equation model with the goal of investigating the different roles played by the hard inputs and soft management employed by institutions in the task of cultivating top-quality innovative doctoral students as well as the specific paths associated with this influence. Our analysis reveals that promoting learning initiative among doctoral students and regulating their level of scientific research engagement are important ways in which various institutional inputs, especially soft management measures, can affect the intrinsic innovation ability of doctoral students, whereas efforts to encourage doctoral students to pursue top-quality academic achievements on the basis of their acquisition of relevant capabilities are related mainly to the hard inputs provided by institutions.

## 1. Introduction

Since the beginning of the 21st century, a new era of technological revolution and industrial transformation, which is characterized by informatization, intelligence, and diversification, has rapidly changed the worldwide divisions of the labor system, which has been in place since the Industrial Revolution. In particular, the development of artificial intelligence (AI) technology, as represented by ChatGPT, has enabled people to realize that the traditional abilities associated with knowledge acquisition, information collation, and solution development are increasingly becoming strong points for computer technologies. Only innovation and creation remain

**Data availability statement:** All relevant data are within the paper and its Supporting information.

**Funding:** This research was supported by the Education Science Planning Project of Jiangsu Province (B/2023/01/205) and the Fundamental Research Funds for the Central Universities of China(SKZZ2025009). Tianbao Zhang is the recipient of the funding awards listed above.

**Competing interests:** No potential conflict of interest was reported by the author.

areas of core competence for human beings that are not susceptible to replacement by machines in the future. The development of creativity in educational contexts is linked to economic and cultural prosperity [1]. An increasing number of countries have embedded an emphasis on developing students' innovation ability in education policy [2–4].

However, the output of innovation ability is not a mechanical motor that can output more power simply when the accelerator is triggered. The linear input–output model cannot effectively explain the mechanism by which innovation capability is generated. This question underlies the assumptions that creativity is an indispensable and complex skill in an increasingly changing world.But creativity can be developed, and that educational contexts are suitable environments for that purpose [5–7]. Student engagement is also widely recognised as an important influence on achievement and learning in higher education and can be researched from a conceptual framework that emerdged four dominant research perspectives: the behavioural perspective, the psychological perspective, the socio-cultural perspective and the holistic perspective [8]. Some European scholars [9] highlight learning environments (activating teaching and learning methods, working life orientation and research, development and innovation (RDI) integration, multidisciplinary learning environments, flexible curricula, entrepreneurship and internationalization) can be associated with students' innovation competences (creativity, critical thinking, initiative, teamwork and networking),the status of a set of crucial soft skills (assertiveness, networking skills, teamwork, sensitivity, socialization, action-orientation, ability to work under pressure and social desirability) in high education [10], self-counciousness competence was an important driver of innovation and environmental commitment in higher education students [11]. In summary, on the basis of education and training, top-quality achievements that surpass most others requires various factors to be combined alongside an adequate organizational platform and a sufficient level of personal cognitive ability. As the highest level of academic education, doctoral education reflects the height of a country's ability to train top innovative talent independently. In the context of globalization and the knowledge economy, doctoral education plays an indispensable role in national efforts and contributes significantly to societal progress and technological development. To cultivate top-quality and innovative doctoral students, a variety of hard inputs are required from the training institution, including relevant funds, facilities, and equipment; an additional requirement in this context is a combination of soft management measures, such as curriculum development, high-quality teaching, interdisciplinary exchange mechanisms, and the establishment of an innovative ecological environment. Therefore, it is necessary to motivate the engagement of doctoral students and to transform the inputs provided by institutions into academic outputs. However, the intermediate link from the inputs provided by institutions to the output of top innovative talent remains largely a cognitive "black box". Although some institutions have invested a large amount of resources in talent training, the results have been insufficient; alternatively, even if good results have been achieved, it has been impossible to explain and summarize the corresponding experience at the theoretical level. By reference to a survey of 1,500 doctoral graduates recruited from throughout

China, as part of this study, the inputs of graduate training institutions are decomposed into two aspects: soft and hard inputs. First, the overall ability the hard inputs and soft management of institutions to improve the intrinsic innovation ability of doctoral students and promote top-quality dissertation is examined. Second, the middle path, which focuses on the academic output of top-quality innovative doctoral students, and the modes of action associated with various input factors are analyzed, thereby providing both theoretical and practical references that can explain the ways in which top-quality innovative talent can be promoted more clearly and cultivate top-quality innovative talent more effectively.

## 2. Current study focus

Since the mid-20th century, institutional studies have represented the greatest form of support for efforts to shift the mode of management and leadership decision-making that characterizes American universities from an "empirical" approach to a "data-driven" model [12]. According to the *IEO* model, the *input–output* process associated with higher education is not linear, and *environmental* factors play an important role in this context. If relevant curricula and teaching are to achieve the expected effects, the educational environment must engage students in terms of their cognition, emotion, and behavior; only in this case can the expected educational output be achieved. Students' participatory learning represents an important channel for the effects of various educational and teaching inputs on academic outputs [13,14].

On this basis, Kuh et al. [15] claimed that one key influence on academic output is the degree to which students engage in educational activities. The actions taken by high-impact colleges and universities to encourage students to engage in long-term, in-depth and comprehensive learning processes can effectively promote students' engagement in educational activities. This effect is cumulative—previous inputs affect subsequent inputs and learning effects on a rolling basis [15,16]. The institutional action model further proposes that the focus of education and teaching research should be on the actions that institutions can effectively take in this regard, not merely on possible influencing factors, with the goal of providing practical guidance for both school and government policies as well as for teachers' efforts; such research should also provide operational guidelines with respect to the educational and teaching behavior exhibited by administrators [17,18].

In terms of efforts to examine the actions taken by the universities in The United States, Europe, etc with regard to graduate training, doctoral education is no longer regarded as an exclusively academic affair but has become an object of institutional management, national policy-making in Europe,and also in China [19]. Baker [20] offer a framework based on three types of fit (person–environment fit, person–culture fit, person–vocation fit) which may provide critical insights into the doctoral student development.Ward [21] extend multidimensional framework of student-doctoral fit and create a more nuanced framework comprising student-doctoral environment fit, student-vocation fit and student-doctoral culture fit, student-(academic)writing fit and student-personal characteristics fit). Three key links between creativity and academic identity formation in doctoral education are 'Becoming a responsible scholar' (connected to social identity), 'Acting with informed agency' (connected to role identity), and 'Expressing the individual voice' (connected to personal identity) [22]. These international literatures provide an effective framework for our research in terms of organizational action and individual participation.Comparatively, the actions taken by Chinese universities with regard to graduate training, Zhang et al. [23] reported that the authorization behavior exhibited by leaders in the context of institutional management has a significant positive effect on team scientific creativity. Gu et al. [24] investigated a sample of physics and chemistry doctoral students who were recruited from the University of Science and Technology of China and reported that the research performance exhibited by students who were recruited from their own school was superior and that the other main influencing factor in this context was the supervisor. However, Luo et al. [25] reported that the educational background of doctoral students had no significant impact on their scientific research performance. In contrast, action measures such as efforts to promote the time inputs of doctoral students and increase their engagement in scientific research, the integration and sharing of cross-disciplinary platforms, the establishment of a graduate supervisory group, the provision of sufficient scientific research resources to students, reformations of the supervisor selection system, improvements in the quality of

the supervisor's scientific research guidance, enhanced curricula, reformed teaching methods, and improvements in the teaching quality of courses can effectively enhance the scientific research performance of doctoral students [25]. Yao and Yu [26] reported that the function of the supervisor must be understood in light of the complete mediating role of graduate students' research self-efficacy in this context and that support provided by fellow students positively moderates the relationship between the support provided by the supervisor and students' own sense of research self-efficacy. Xie and Wang [27] investigated an academic research group, revealing that the role played by supervisors in the task of cultivating doctoral students mainly involves stimulating the academic interests and abilities of doctoral students and serving as a spiritually inspiring leader and supporter of students' actions, particularly by encouraging them to enhance their personal and academic qualities. The supporters of education coordinate various stimulating factors according to their aptitudes and attach a great deal of importance to the task of guiding junior students' academic direction and senior students' career planning [27]. Li and Zhang [28] investigated efforts to promote engagement among science and engineering graduate students in the context of these students' scientific research practice. These authors revealed that a higher degree of diversity with respect to the engagement of scientific research practice can facilitate the direct improvement of students' innovation ability; namely, the more deeply students engage in scientific research practice, the higher levels and degrees of systematicity that they obtain, and the more conducive this approach is to the task of cultivating students' innovation ability [28]. Li et al. [29] also focused on the role played by scientific research engagement in this context. These authors revealed that the significant positive effect of the number of projects on the scientific research ability of doctoral students is evident only in the fields of natural sciences and social sciences, i.e., not in the humanities. The significantly positive impact of the quality of project engagement on the value added with respect to doctoral students' scientific research capabilities has been supported in all the natural sciences, social sciences, and humanities; thus, the impact of the quality of project engagement on the value added with respect to doctoral students' scientific research capabilities is stronger than that of the amount of project engagement [29]. Bai and Huang [30] reported that clear doctoral degree requirements have the strongest influence on academic innovation, whereas financial aid factors have only a small influence in this regard; furthermore, the process variables of degree requirements, learning methods and teacher–student relationships lie at the core of these influences. As the most important factor with regard to doctoral students' academic innovation ability, the indirect influence effect of these students' academic motivation on their academic innovation ability is relatively large; in particular, the academic motivation→degree requirement→academic innovation ability path is associated with a typical mediating effect in this context [30]. However, an analysis of data drawn from the National Survey of Doctoral Graduates, which was conducted by Li [31], contradicted the claim that stricter publication requirements for graduation dissertations are superior. The relationship between the number of papers published by doctoral students in the natural sciences and the added value with respect to their scientific research capabilities exhibited an inverted U-shaped curve. Thus, we should understand the educational value of the publication of academic dissertations by doctoral students correctly. Schools and supervisors should correct the previous erroneous perspective that "publication is everything" [31].

A relatively consistent conclusion drawn by the studies referenced above is that training institutions and supervisors should invest their best efforts in measures aimed at stimulating the motivation, interest, aspiration and active learning behavior of students with the goal of promoting their learning engagement and thus transforming the institution's action inputs into academic outputs. However, many inconsistent views have been expressed in this regard. For example, in terms of student selection strategies, no conclusive evidence has yet been found to support the conclusion that better student sources are more conducive to efforts to improve the innovation ability of doctoral students. As a result, the current evidence-based foundation on which the training strategies used in graduate schools rely is insufficient. Important reasons for this situation include the fact that the scale at which many studies on this topic have focused their investigations is small, the samples used in such research have been insufficiently representative, and analyses of the connections, relationships and action paths among the relevant variables pertaining to the process of cultivating top innovative talent have been inadequate.

## 3. Method and research hypothesis

### 3.1. Research sample

This study strictly adhered to the standards and guidelines of ethical research and received approval from the Ethics Committee of Nanjing Agriculture University.Between mid-May 2021 and early July 2021, this study commissioned a professional research institution to conduct a nationwide survey among doctoral students who graduated in that year or were within two years of graduation. The vast majority of the respondents were at the graduating stage of doctoral education. This study was distributed online. To alleviate respondents' concerns and enhance the validity of the questionnaires, this study adopted anonymous and sensitive wording in the questionnaire design. Before the commencement of data collection, all potential participants were clearly informed about the purpose and intent of the research. No parents or guardians were contacted in the consent process since all participants were 18 or older.

The sampling process was as follows. This research focused on four dimensions: the resident population (as indicated by the number of groups covered), economic development (as indicated by the quality of the local internet infrastructure), educational resources (as indicated by the quality of netizens) and the activity level of netizens (as indicated by the feasibility of administering the online survey). Different weights were assigned to the observations obtained in prefecture-level cities with respect to each dimension. According to the comprehensive score as well as the balanced distribution of colleges and universities within each region, first-tier and second-tier universities had more resources when they were located in North China or East China; therefore, more observations were collected in these contexts, although the study aimed to cover all levels of cities across the country. Following the data cleaning process, a total of 1,500 valid samples were obtained, which included 238 universities, including 695 observations obtained from world-class universities, 356 observations obtained from universities featuring world-class disciplines, and 449 samples obtained from other universities(Table 1). The sample included 1006 male participants, accounting for 67.1% of the total, and 494 female participants, accounting for 32.9% of the total. The gender ratio is 5.1 percentage points lower than the proportion of female PhD students reported in the China Education Statistical Yearbook 2020. While the proportion in the Statistics Yearbook represents the status of all students in that year, rather than the male-to-female ratio of graduates. The longitudinal percentage of female PhD students from 2015 to 2020 (37.85%, 38.63%, 39.27%, 40.37%, 41.32%, 41.87%, respectively) exhibits an upward trajectory. The 32.9% representation of women in the survey sample basically aligns with the average value 39.88% of PhD students in this sample during this period.

Engineering graduates accounted for the highest proportion of participants (25.7%), followed by the fields of science (15.4%), economics (14.4%), medicine (9.3%), management (8.5%), education (7.0%), law (5.8%), literature (5.3%), agronomy (3.0%), philosophy (2.7%), history (2.0%), and art (0.8%). In comparison to the discipline distribution of PhD graduates in China as reported in the China Education Statistical Yearbook 2020, samples from economics, management, and education is marginally elevated, while samples from science, engineering, and medicine is slightly diminished, with the proportion of other disciplines remaining relatively consistent. This occurrence is likely attributable to the different willingness of students to engage in social research initiatives from various disciplines. Previous data monitoring indicates that, as a non-mandatory research initiative, the willingness to respond and actual involvement rates of PhD students are markedly lower major in science, engineering, and medicine than whom major in the humanities and social sciences. These proportions are basically identical to the corresponding proportions among all doctoral students in China.

Based on the results of ANOVA statistical analysis, there were no significant inter-group differences in gender (Table 2) and disciplines (Table 3), it indicate that all samples were essentially homogeneous.

### 3.2. Definitions and descriptive statistics concerning top-quality innovative doctoral students

This study explores top-quality innovative doctoral students in terms of two aspects. First, these students exhibit intrinsic innovative ability; second, on the basis of such ability, their actual academic performance surpasses that of the majority

**Table 1. Sample distribution.**

| | | Number | Proportion |
|---|---|---|---|
| Gender | Male | 1006 | 67.1 |
| | Female | 494 | 32.9 |
| Location of family | Metropolitan | 883 | 58.9 |
| | Prefecture-level city | 408 | 27.2 |
| | County-level city | 158 | 10.5 |
| | Town | 25 | 1.7 |
| | Rural | 26 | 1.7 |
| Type of university | World-class universities | 695 | 46.3 |
| | Universities featuring world-class disciplines | 356 | 23.7 |
| | Other universities | 449 | 29.9 |
| Discipline | Philosophy | 41 | 2.7 |
| | Economics | 216 | 14.4 |
| | Law | 87 | 5.8 |
| | Education | 105 | 7.0 |
| | Literature | 80 | 5.3 |
| | History | 30 | 2.0 |
| | Science | 231 | 15.4 |
| | Engineering | 385 | 25.7 |
| | Agronomy | 45 | 3.0 |
| | Medicine | 140 | 9.3 |
| | Management | 128 | 8.5 |
| | Art | 12 | 0.8 |

**Table 2. ANOVA in gender.**

| | | Sum of Squares | df | Mean Square | F | Sig. |
|---|---|---|---|---|---|---|
| Creativity | Between Groups | .000 | 1 | .000 | .000 | .997 |
| | Within Groups | 321.290 | 1498 | .214 | | |
| | Total | 321.290 | 1499 | | | |
| Excellent Dissertations | Between Groups | .037 | 1 | .037 | .315 | .574 |
| | Within Groups | 176.946 | 1498 | .118 | | |
| | Total | 176.983 | 1499 | | | |

**Table 3. ANOVA in discipline.**

| | | Sum of Squares | df | Mean Square | F | Sig. |
|---|---|---|---|---|---|---|
| Creativity | Between Groups | 1.446 | 6 | .241 | 1.125 | .345 |
| | Within Groups | 319.845 | 1493 | .214 | | |
| | Total | 321.290 | 1499 | | | |
| Excellent Dissertations | Between Groups | .276 | 6 | .046 | .388 | .887 |
| | Within Groups | 176.708 | 1493 | .118 | | |
| | Total | 176.983 | 1499 | | | |

 

of doctoral students. On the basis of this operational definition, the top-quality innovative doctoral students included in the sample investigated in this research were measured and identified by reference to two indicators. First, the Williams Creativity Scale was used to measure the innovation ability exhibited by doctoral students. Second, top-quality academic innovation is indicated when students' dissertations at the time of their graduation are assessed as excellent. As the final outcome of doctoral students' studies and research, these dissertations comprehensively reflect their scientific research ability, innovation ability, ability to grasp and apply knowledge, and written expression ability [32]. Moreover, the level of innovation that characterizes such a dissertation represents a concentrated reflection of doctoral students' academic ability and stands as the best proof of their academic achievement [33].

The statistics calculated for this research reveal that among the 1,500 participants, 205 dissertations were rated as excellent, accounting for 13.67% of the total. Among the authors of such excellent dissertations, men accounted for 14.02% of the total, i.e., slightly higher than the proportion of women (12.96%). Excellent dissertations accounted for 16.12% of dissertations at world-class universities, 14.04% of dissertations at universities featuring world-class disciplines, and 9.58% of dissertations at other universities, thus indicating a decreasing trend. The proportions of excellent dissertations in the humanities and social sciences (including the seven disciplines of literature, history, philosophy, law, economics, management and art) and in science and engineering (including the four disciplines of science, engineering, agriculture and medicine) accounted for 14.02% and 14.02% of the total, respectively. These proportions were essentially identical (13.36%) (Table 4).

The Williams Creativity Scale is an innovation performance scale that has been widely used worldwide. This study retained the structure of this scale and revised the item descriptions in light of the characteristics of doctoral students and the actual situation of their education in China (Table 5). The items were scored on a seven-point Likert scale, in which context options 1–7 indicated the degree to which the behavior of the survey respondents was in line with the various items,

Table 4. Proportion of excellent doctoral dissertations (%).

| | Gender | | Type of university | | | Discipline | |
|---|---|---|---|---|---|---|---|
| | Men | Women | World-class universities | Universities featuring world-class disciplines | Other colleges and universities | Literature, history, philosophy, law, economics, management, and art | Engineering, agriculture and medicine |
| Other papers | 85.98 | 87.04 | 83.88 | 85.96 | 90.42 | 85.98 | 86.64 |
| Excellent dissertations | 14.02 | 12.96 | 16.12 | 14.04 | 9.58 | 14.02 | 13.36 |

Table 5. Sample items and descriptive statistics concerning the measurement of the innovation ability of doctoral students.

| Primary dimension | Secondary dimension | Reference measurement tool | Number of items | Example measurement items | Overall mean | Excellent mean | Other mean | Analysis of variance (ANOVA) significance |
|---|---|---|---|---|---|---|---|---|
| Creativity | Adventurousness | Williams Creativity Scale | 4 | I do not like excessive rules and restrictions | 4.21 | 4.29 | 4.19 | 0.008 |
| | Curiosity | | 4 | I like to explore the reasons for which things happen | 4.91 | 5.01 | 4.89 | 0.001 |
| | Imagination | | 4 | When I read a novel or watch television, I like to think of myself as a character in the story | 5.31 | 5.41 | 5.29 | 0.065 |
| | Challenging | | 4 | When performing research, I prefer not to use conventional research methods | 5.49 | 5.70 | 5.46 | 0.000 |
| | Overall | | 16 | | 4.98 | 5.10 | 4.96 | 0.000 |

ranging from not at all consistent to completely consistent. A score of 4 indicated an intermediate state. A comprehensive average score between 6 and 7 indicated that the survey respondents exhibited strong positive innovation ability. A score of 5 indicated that the survey respondents exhibited positive innovation ability. A score of 4 indicated that the survey respondents exhibited neutral innovation ability. Finally, a score of less than 4 indicated that the survey respondents exhibited conservative tendencies. The Cronbach's α coefficient for this scale was 0.701, thus indicating good reliability. After performing a Kaiser-Meyer-Olkin (KMO) test and Bartlett's test of sphericity on the scale, we conducted a confirmatory factor analysis. The results revealed that the number and structure of the principal components were completely consistent with the dimensions included in the scale, thus indicating good validity. The survey results revealed that the overall mean creativity of the samples in this survey was 4.98, which was close to the level indicating strong innovation ability according to the criteria used to design the scale; the standard deviation of individual creativity scores was 0.46, and the scores of the middle 50% of the samples ranged between 4.75 and 5.31. The distribution at the individual level was more concentrated. In terms of internal structure, among the four secondary dimensions, the mean value of challenge was only 5.49. The second highest value pertained to imagination, which exhibited a mean value of 5.31. The mean value of curiosity was 4.91. Finally, the mean value of adventurousness was the lowest, i.e., 4.21. To compare the mean values for innovation ability between the doctoral students who produced excellent doctoral dissertations and other doctoral students, an analysis of variance (ANOVA) was conducted; the results revealed that the group of doctoral students who produced excellent dissertations obtained significantly higher scores than did the other group of students in both the primary and secondary dimensions.

### 3.3. Research hypothesis

The overall aim of the study is, first, to investigate the influences of different modes and levels of institutional input on the intrinsic innovation ability and realistic innovation academic performance of doctoral students by conducting regression analyses while controlling for personal, family and school characteristics. On the basis of the extant literature on this topic, the following two specific research hypotheses are proposed:

H1: Increasing institutional input can enhance the intrinsic innovation ability and academic performance of doctoral students in terms of realistic innovation.

H2: The roles played by the hard inputs and soft management associated with training institutions in the task of promoting the growth of top-quality innovative doctoral students may differ.

In addition, according to modern institutional research theory, the effect of institutional inputs on students' academic output is often indirect. For example, although fund inputs are helpful with respect to students' academic performance, funds cannot purchase excellent dissertations directly. A balance between the inputs and the outputs should be established. Various intermediate links are also relevant in this context. For top-quality innovative doctoral students, their own behaviors, such as active learning, knowledge construction, and critical creation, represent direct influences on their abilities and academic performance. If institutional input can motivate and regulate students' learning behaviors, such as those pertaining to tracking academic trends and interdisciplinary exploration, it can also encourage them to develop innovation ability and exhibit academic excellence. Therefore, a third research hypothesis is proposed as follows:

H3: Stimulating and regulating student engagement are important ways in which institutional input can play a role in this context.

### 3.4. Measurement of the main explanatory variables

On the basis of a comprehensive consideration of the action theories on which foreign institutions rely as well as the action practices employed by graduate students in China, in the present study, institutional inputs pertaining to the training

of top-quality innovative doctoral students are measured in terms of 10 aspects, in which context hard inputs include scientific research funding, opportunities to engage in academic exchange, study space, study materials, and scholarships and grants. Soft management includes school management, professional basic courses, methodological courses, academic frontier courses, and interdisciplinary courses. In addition, school type (i.e., first-class universities, universities featuring first-class disciplines, and other universities) is included as an indicator of the level of comprehensive input. The measurement of student engagement focused on four aspects, i.e., academic exchange, academic trend tracking, classroom input, and interdisciplinary exploration; each aspect was associated with a number of specific behaviors (see Table 6 for the specific indicators and explanations).

## 4. The effect of institutional training input on the cultivation of top-quality innovative doctoral students

### 4.1. Direct impact

In light of the results of the multivariate linear regression performed to investigate the intrinsic innovation ability of doctoral students (see the 3rd and 4th columns of Table 7 for further details), 9 of the 10 institutional educational inputs under investigation exhibited a significant positive promoting effect. In terms of hard inputs, for every 1 standard deviation increase in the level of scientific research funds, study spaces, study materials, and scholarships provided as inputs, the intrinsic innovation ability of doctoral students increases by 0.152, 0.072, 0.06, and 0.075 standard deviations, respectively. In terms of soft management, for every 1 standard deviation increase in the level of school management or the richness and effectiveness of professional basic courses, methodological courses, academic frontier courses, and interdisciplinary courses, the intrinsic innovation ability level of doctoral students increases by 0.101, 0.055, 0.124, 0.147, and 0.074 standard deviations, respectively. The regression coefficient of the institution type, which represents the comprehensive input level, is not significant; that is, no significant differences are observed among first-class universities, universities featuring first-class disciplines and other universities. Because we used standard regression coefficients, we were able to compare the effect sizes of inputs from different institutions on the intrinsic innovation ability of doctoral students on the basis of the same 1 standard deviation increase. In general, the most influential institutional input factor is sufficient research funding, followed by the provision of sufficiently rich and effective academic frontier courses and methodological courses, the school's management level and various forms of scholarships and grants at sufficient amounts, sufficient interdisciplinary courses, sufficient study spaces, such as self-study rooms, and learning materials such as books and periodicals. The provision of sufficiently rich and effective professional basic courses has relatively little influence in this context.

The results of a multivariate logistic regression on the top-quality innovation and academic performance of doctoral students (Table 7, Columns 5 and 6) revealed that none of the five indicators of soft management on the part of the training institution had a significant effect on the probability that the doctoral dissertation would be evaluated as excellent. In terms of direct impacts, with regard to hard inputs, the provision of opportunities to engage in academic exchange and scholarships did not have significant effects in this context. Only the provision of scientific research funding and support alongside sufficient study materials and study spaces was revealed to have a significant positive promoting effect is this context. When the levels of such inputs increase by 1 standard deviation, the probability of the dissertation being rated as excellent increases by 0.642, 0.618 and 0.588 standard deviations, respectively. Once again, no significant effect of the comprehensive input was observed.

In general, although the various hard inputs and soft management measures associated with graduate training institutions have significantly improved the intrinsic innovation ability of doctoral students directly, on the basis of the training of their abilities, only hard inputs, such as scientific research funding and support as well as the provision of sufficient study materials and study space, can influence the likelihood of students' dissertations being evaluated as excellent, while soft management measures, such as school management, were not observed to have a significant direct promoting effect in this context.

**Table 6. Measurements of institutional input and student engagement.**

| Primary indicators | Secondary indicators | Indicator description |
|---|---|---|
| Hard input | Research funding | The school provides sufficient research funding and support |
| | Opportunities to engage in academic exchange | The school provides ample opportunities to engage in academic exchange |
| | Study spaces | The school has sufficient learning spaces, such as study rooms |
| | Study materials | The school has sufficient books, periodicals and materials |
| | Scholarships and grants | The school provides various forms of scholarships at sufficient amounts |
| Soft management | School management | I am satisfied with the school's level of management |
| | Professional basic courses | The professional basic courses offered by the school are rich and effective |
| | Methodological courses | The methodological courses offered by the school are rich and effective |
| | Academic frontier courses | The academic frontier courses offered by the school are rich and effective |
| | Interdisciplinary courses | The school offers a sufficient number of interdisciplinary courses |
| Comprehensive input | Types of doctoral schools | First-class universities, universities featuring first-class disciplines, and other universities |
| Student engagement | Academic exchanges | Attend academic lectures |
| | | Participate in domestic academic conferences |
| | Academic trend tracking | Read the most up-to-date papers in international journals |
| | | Listen to academic reports delivered by international high-level experts |
| | | Understand the most up-to-date scholarly advancements on the basis of discussions with classmates |
| | Classroom input | Ask insightful questions in the classroom |
| | | Evaluate the views of others carefully in the classroom |
| | | Share one's own academic understanding in class discussions |
| | Interdisciplinary exploration | Read relevant academic materials pertaining to other disciplines |
| | | Listen to interdisciplinary reports or participate in courses offered by other departments |
| | | Communicate with teachers and classmates from other disciplines |
| | | Focus on the research methods used in other disciplines |
| | | Be able to integrate knowledge obtained from other courses into class discussions |

## 4.2. Modes of influence

With regard to H3, 13 types of learning and scientific research behaviors on the part of doctoral students were observed in this research. These behaviors can be divided into four categories, i.e., academic communication, academic trend tracking, classroom input, and interdisciplinary exploration, which are used as intermediate variables to analyze institutional inputs, thereby opening the "black box" of academic outputs. Because multiple explanatory variables and intermediate variables are involved in this process simultaneously, the influencing coefficients are identified on the basis of a

**Table 7. Results of the regression analysis of institutional input on the intrinsic innovation ability and realistic innovation performance of doctoral students.**

| Independent variable | Dependent variable Regression methods | Intrinsic ability (creativity) | | Realistic performance (excellent dissertations) | |
|---|---|---|---|---|---|
| | | Multiple linear regression | | Multivariate logistic regression | |
| | | Standardized regression coefficient | Standard deviation | Standardized regression coefficient | Standard deviation |
| Hard input | Research funding | 0.152*** | (0.010) | 0.642* | (0.085) |
| | Opportunities to engage in academic exchange | 0.030 | (0.011) | 0.045 | (0.095) |
| | Study spaces | 0.072** | (0.010) | 0.588* | (0.093) |
| | Study materials | 0.060* | (0.011) | 0.618* | (0.093) |
| | Scholarships and grants | 0.075** | (0.011) | 0.577 | (0.094) |
| Soft management | School management | 0.101*** | (0.012) | 0.241 | (0.108) |
| | Professional basic courses | 0.055* | (0.011) | 0.227 | (0.098) |
| | Methodological courses | 0.124*** | (0.011) | −0.033 | (0.095) |
| | Academic frontier courses | 0.147*** | (0.011) | −0.307 | (0.094) |
| | Interdisciplinary courses | 0.074** | (0.010) | −0.596 | (0.087) |
| Comprehensive input (other universities are used as the reference) | First-class universities | −0.071 | (0.035) | −0.401 | (0.311) |
| | Universities featuring first-class disciplines | 0.014 | (0.034) | −0.031 | (0.302) |
| Control Variable | Personal characteristics (i.e., gender, party membership, motivation, and major), family characteristics (i.e., family economic capital, family cultural capital, family social capital, and type of permanent residence), institution characteristics (i.e., type of school at which the master's degree was obtained, type of school at which the undergraduate degree was obtained, educational system, admission methods, and training methods) | | | | |
| Intercept | | 3.040 | | 0.004 | |
| Sample size | | 1500 | | 1500 | |
| Pseudo $R^2$/$R^2$ | | 0.389 | | 0.093 | |
| LR chi²(45)/F | | 20.1 | | 111.32 | |

Notes: 1. The values reported in this table are standardized regression coefficients; 2. The meanings of the significance markers are as follows: * $p < 0.05$, ** $p < 0.01$, and *** $p < 0.011$.

structural equation model regression. The results (which are illustrated in the right half of Fig 1) reveal that the academic exchanges, academic trend tracking, classroom input and interdisciplinary exploration behavior on the part of doctoral students all have significant positive effects on doctoral students' intrinsic innovation ability, in which context classroom input has the strongest influence, followed by interdisciplinary exploration and academic trend tracking; in contrast, academic communication has the weakest influence. These findings indicate that, in terms of learning behavior, when doctoral students ask insightful questions in class, evaluate the views of others carefully, and share their own academic understanding through class discussions, these behaviors have a direct promoting effect on doctoral students' intrinsic innovation ability. Reading relevant academic materials pertaining to other disciplines, listening to interdisciplinary reports or participating in courses offered by other departments, communicating with teachers and classmates in other disciplines, paying attention to the research methods used in other disciplines, being able to integrate knowledge obtained from other courses into class discussions, exploration behavior, and behaviors that enable students to follow academic trends, such

as reading the most up-to-date papers in international journals, listening to academic reports delivered by high-level international experts, and understanding the most up-to-date scholarly advancements on the basis of discussions with classmates, are also important approaches that can affect intrinsic innovation.

Accordingly, which institutional inputs, as direct effects on innovation ability, can motivate and regulate learning and scientific research behaviors among these students? The results of the regression indicate the following (as presented in the left half of Fig 1). ① Five inputs, namely, the provision of sufficient research funding, scholarships and grants, sufficiently rich and effective methods, academic frontier courses and interdisciplinary courses, all had significant stimulating and regulatory effects on the four types of learning and scientific research behaviors exhibited by doctoral students. ② Two institutional inputs can affect the three types of study and scientific research behaviors exhibited by doctoral students. Specifically, the level of school management has a promoting effect on academic exchanges, classroom inputs and interdisciplinary exploration among doctoral students. The provision of sufficiently rich and effective professional basic courses has a promoting effect with regard to doctoral students' tracking of academic trends, classroom inputs, and interdisciplinary exploration ③ The institutional input that can affect two types of learning and scientific research behaviors is the provision of sufficient study materials, which can encourage doctoral students to track academic trends and influence the level of classroom inputs. ④ The institutional input that can affect one type of learning and scientific research behavior

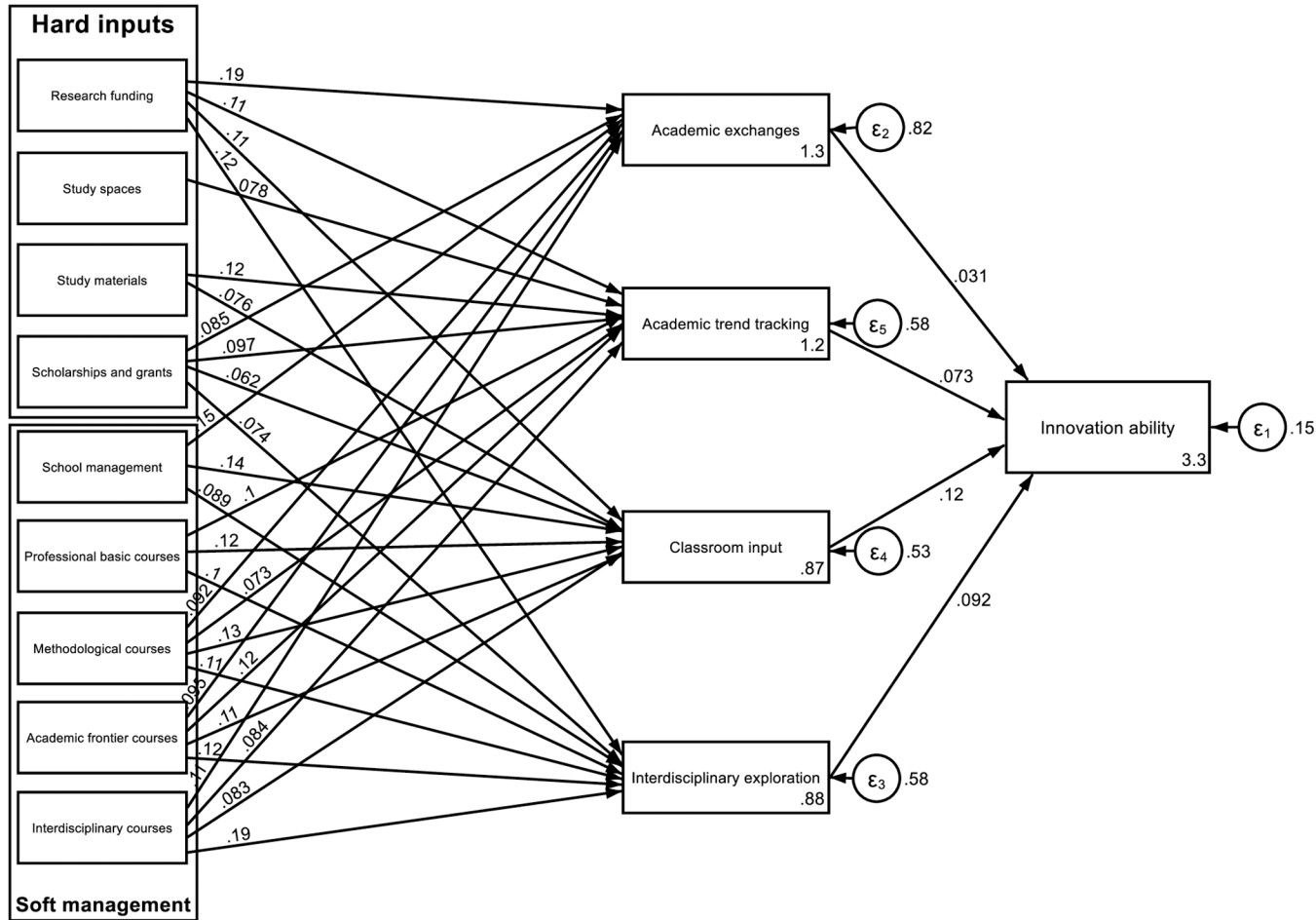

**Fig 1. Paths that institutions can take to invest in the efforts to promote the intrinsic innovation ability of doctoral students.**

is the provision of sufficient study spaces, which can significantly promote only academic trend tracking behavior among doctoral students.

Furthermore, the influence of institutional input on students' academic performance in terms of top-quality innovations is analyzed. The regression results (Fig 2) reveal that, among a large number of learning and scientific research behaviors on the part of doctoral students, five factors can have significant positive impacts on the likelihood of their dissertations being assessed as excellent, including studying abroad, being independently responsible for scientific research projects, obtaining sufficient research funding, reading the most up-to-date international papers and attending academic conferences. These results are in line with the relevant conclusions of numerous studies on this topic. The focus of the present research is not on the effects of these factors on the quality of dissertations but rather on which institutional inputs can stimulate and regulate these factors. The results presented in the structural equation modeling analysis support the following conclusions: ①Although the type of institution does not directly determine whether a given dissertation will be evaluated as excellent, it can directly affect doctoral students' efforts to study abroad and thus indirectly affect the quality of their dissertations, thus reflecting the practical significance of the ability of first-class institutions to serve as platforms for the academic output of doctoral students. ② Research funding significantly affects the degree to which doctoral students are independently responsible for scientific research projects, thus enabling them to obtain sufficient research funding and read the most up-to-date international papers; this factor thus indirectly increases the quality of dissertations. This influence reflects the important role played by scientific research funding in efforts to promote innovation with regard to doctoral students' academic outputs. ③ With respect to the same input in terms of funds, the provision of various forms of scholarships at sufficient amounts can also affect the research funding received by doctoral students, their ability to read of the most up-to-date international papers, and their engagement in academic conferences, thus indicating that after

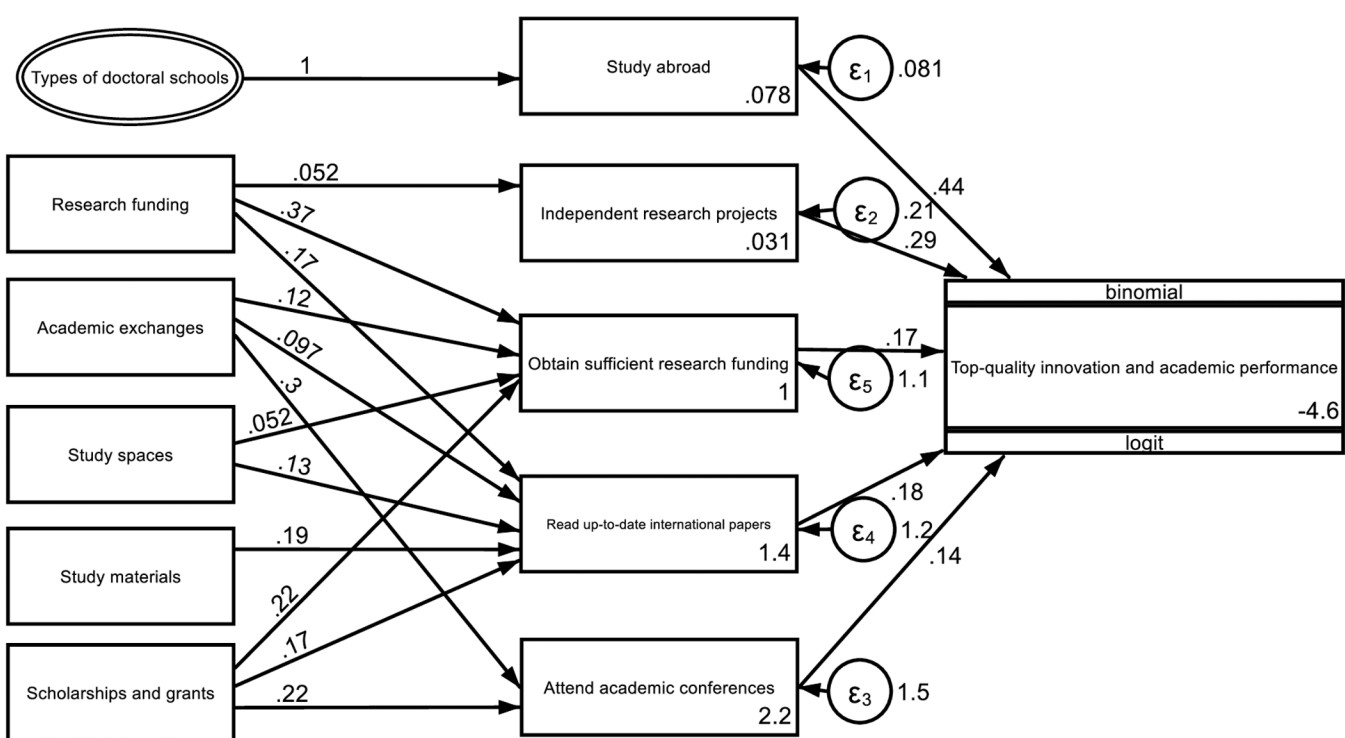

**Fig 2. Paths that institutions can take to invest in efforts to promote the academic performance of doctoral students in terms of top-quality innovation.**

doctoral students receive scholarships, many of them may use these funds to support their studies rather than their personal lives. ④ The ability of the school to provide opportunities to engage in academic exchange is reflected in the ways in which it helps doctoral students to attend academic conferences, read the most up-to-date international papers and obtain research funding; in which context their participation in academic conferences and reading of the most up-to-date papers represent the original input intentions. Such help for doctoral students receiving research funding represents an unexpected finding of this research, which may reflect the fact that increased opportunities to engage in academic exchanges can improve the success rate of doctoral students' applications for funding for their research projects. ⑤ In line with our expectations, in terms of school infrastructure input, the provision of sufficient study materials, such as books and journals, helped doctoral students read the most up-to-date international dissertations, although this effect was limited to that context; in addition, the provision of sufficient study spaces, such as study rooms, helped doctoral students. In addition to enabling students read the most up-to-date papers, such spaces can affect their ability to acquire funding for scientific research, thus indicating that for doctoral students, the role of such study spaces does not focus solely on reading and studying materials but also on conducting scientific research.

## 5. Conclusion and discussion

### 5.1. Conclusion

The cultivation of top-quality innovative doctoral students is impacted by many factors. This study analyzes the role played by institutional input in efforts to increase the academic performance of top-quality innovative doctoral students, particularly with respect to the intermediate process and specific mode of action leading from input to output. On the basis of an investigation of 1,500 graduating doctoral students, after controlling for personal, family, and institutional characteristics (especially the characteristics of the schools at which students obtained their master's and undergraduate degrees), the following findings were obtained.

The areas in which hard inputs and soft management play particular roles in doctoral training institutions differ. Soft management inputs, such as the school's level of management, curriculum, and teaching methods, significantly affect the intrinsic innovation ability exhibited by doctoral students but are not significantly correlated with the achievement of excellent doctoral dissertations, whereas hard inputs, such as funds, study spaces and study materials, significantly increase the intrinsic innovation ability of doctoral students and promote the actual top-quality innovation academic performance that is associated with excellent dissertations. Accordingly, the soft management measures employed by the school mainly impact the intrinsic ability of doctoral students. On the basis of doctoral students' development of innovation ability, the achievement of top-quality academic performance depends on the hard inputs provided by training institutions. Currently, there is no research directly indicating the intrinsic mechanism of hard inputs on the top innovative performance of doctoral candidates. An ambidextrous learning in organizational learning research provides a corresponding explanation [34]. Based on how cognition and context are tightly bound, organizational learning theory identifies two different logics of learning: executional learning [35], which develops into the concept of exploitative learning [36], and developmental learning [35], which evolves into the concept of exploratory learning [37]. These two logics of learning as a duality are contradictory in nature but mutually enabling and interdependent as a potential ambidextrous learning process. The results of this study may show that the soft management of the university is more correlated with the executional learning or exploitative learning of doctoral students, which emphasizes the integration and application of existing resources, while the hard investment is more correlated with the developmental learning or exploratory learning of doctoral students, which focuses on the discovery and innovation of new resources.

Stimulating and regulating the learning and scientific research engagement behavior of doctoral students are important ways in which institutional input can play a role in this context. Students' learning initiative and engagement in scientific research, which affect students' development of relevant abilities directly, largely determine the role played by institutional

training input in the process that links resource inputs with academic outputs. The analysis of the breadth of the institution's stimulation of inputs and regulation of the learning and scientific research engagement behavior exhibited by doctoral students, which was conducted on the basis of the structural equation model, reveals that the provision of sufficient research funding and scholarships had the strongest effect and significantly stimulated and regulated all kinds of learning and scientific research behaviors on the part of doctoral students. The funding of doctoral students through research assistantships significantly improves the research participation of doctoral students, and they are more likely to obtain higher academic research achievements [38]. Offering sufficiently rich and effective methodological courses, academic frontier courses and interdisciplinary courses as well as implementing soft measures, such as by providing a high level of campus management, can help promote academic exchanges, academic trend tracking, classroom input and interdisciplinary exploration behavior among doctoral students. Private study materials and study spaces can promote learning behaviors such as reading the most up-to-date papers to a certain extent and can make doctoral students' engagement in scientific research more convenient. A vibrant academic atmosphere and abundant resources become fertile ground for nurturing innovative thinking, inspiring doctoral students to actively engage in academic exploration [39]. Although the type of institution reflects the factor of comprehensive input ability, the only significant direct effect observed in this context lies in the ability of the type of institution to facilitate study abroad. Studying abroad increases doctoral students' engagement with the frontiers of international research, which in turn promotes academic performance.

## 5.2. Discussion

To improve the quality of doctoral training and cultivate top innovative talent, doctoral training institutions in many contries,including China have increased inputs in all respects over the years. On the basis of the novel findings revealed by this study, we propose the following suggestions:

1. First of all,a relatively independent space for study and scientific research is needed to guarante. In our study,due to objective limitations imposed by basic conditions on campus or subjective ignorance of the importance of such independent spaces for doctoral students, a certain number of doctoral training institutions in China can't provide a sufficient number of relatively independent spaces for doctoral students.In general, an independent space can protect doctoral students from interference; establish an environment that is conducive to study and research; promote better thinking, exploration and innovation; ensure a focus on experimenting and writing; and improve work efficiency. This study also revealed that providing sufficient study spaces, such as self-study rooms, can help doctoral students track academic trends and obtain funding and other forms of support for scientific research projects. With regard to efforts to guarantee the independence, autonomy and participation of doctoral students, training institutions should provide relatively independent study and scientific research spaces for doctoral students as the most basic hard inputs,which a better student engagement and a better soft environment all depend on.

2. The provision of sufficient financial support. Funding and support constitute an important foundation for innovative scientific research activities. This study reveals that this claim is also applicable to doctoral students. The provision of sufficient research funds and scholarships by training institutions can significantly stimulate and regulate scientific research among doctoral students, which is helpful with respect to their learning and scientific research behaviors, which enable them to develop form intrinsic innovation ability and produce excellent dissertations. Doctoral schools should be aware of the importance of financial support with respect to the training of top and innovative doctoral students, and while these schools provide guaranteed scientific research funds for teachers, they should also provide special guarantees for scientific research funds for students. In addition, it should be noted that the finding of this study indicating that students use the doctoral scholarships they receive to fund their learning and scientific research behavior to a considerable extent, rather than spending it solely on their personal lives and consumption, should be rooted in a comprehensive consideration of the funding structure. Because the uses of special scientific research funds are

relatively rigid, whereas the uses of scholarships and grants are more flexible, the proportion of grants and funds provided to doctoral students can be increased appropriately. In particular, research assistants and teaching assistants are suggested as the primary funding types [38].

3. It is important to pay attention to the tasks of establishing a soft campus environment and cultivating an innovative education ecology. The development of innovation ability is a complex process that requires individuals to acquire relevant professional knowledge and skills as well as to develop their personalities and minds. Only when individuals cognitive and noncognitive abilities are integrated can they develop higher orders of innovation ability. In this process, doctoral students must select courses, undergo examinations, produce research proposals, and write and defend dissertations, which are influenced by various factors in the institutional environment. The results of the present study indicate that the level of school management has significant stimulating and moderating effects on academic exchange, classroom input and interdisciplinary exploration among doctoral students, thus leading to improvements in their innovation ability. In summary, through the implementation of comprehensive strategies, the goal is to establish a healthy academic ecosystem that encourages both knowledge exploration and application, fosters innovation and collaboration, and provides solid support for doctoral students' academic pursuits and career development. This approach can increase doctoral students'research engagement and facilitate their development of innovation ability.

### 5.3. Limitations and future research

While this study employed a multi-group data collection method to reduce common method bias, its reliance on self-reported data from doctoral students may still be subject to some degree of common source bias. To further validate the findings of this study, future research could consider adopting experimental or quasi-experimental designs, or incorporating objective data sources, to enhance the accuracy and reliability of the results. Additionally, this research primarily explores the impact of institutional training input on the cultivation of top-quality innovative from the perspective of doctoral students, without fully considering the potential roles of internal and external factors such as personal,social or economic. Factors such as the personal psychology, changing guidelines and geopolitical influences may also influence doctoral students' research engagement. Furthmore, the definition of the top-quality innovative in academic performance is a little narrow from an academic perspective. It needs a broader and more scientific definition. Lastly, given that the sample of this study is limited to doctoral students in China, the cultural universality of its conclusions remains to be further verified. Hence, subsequent research should consider cross-cultural institutional training input on the cultivation of top-quality Innovative to explore and verify the universality and specificity of the findings of this study.

### Supporting information

**S1 File. Underlying data.**
(XLSX)

### Author contributions

**Conceptualization:** Zhimin Liu.

**Investigation:** Weihai Huang.

**Methodology:** Weihai Huang.

**Supervision:** Zhimin Liu.

**Writing – original draft:** Tianbao Zhang.

**Writing – review & editing:** Tianbao Zhang.

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
