## [Decision Letter · Decision Letter 0]

Dear Dr. Zhang,

Thank you for submitting your manuscript to PLOS ONE. After careful consideration, we feel that it has merit but does not fully meet PLOS ONE’s publication criteria as it currently stands. Therefore, we invite you to submit a revised version of the manuscript that addresses the points raised during the review process.

We look forward to receiving your revised manuscript.

Kind regards,

Claudia Noemi González Brambila, Ph.D.

Academic Editor

PLOS ONE

Journal Requirements:

1. Please ensure that your manuscript meets PLOS ONE's style requirements, including those for file naming. The PLOS ONE style templates can be found at https://journals.plos.org/plosone/s/file?id=wjVg/PLOSOne_formatting_sample_main_body.pdf and https://journals.plos.org/plosone/s/file?id=ba62/PLOSOne_formatting_sample_title_authors_affiliations.pdf .

3. In the online submission form, you indicated that [The data that support the fndings of this study are available from the corresponding author, upon reasonable request.].

4. Please include a caption for figure 1, 2 and table 1, 2, 3, and 4.

Reviewers' comments:

Reviewer's Responses to Questions

**Comments to the Author**

1. Is the manuscript technically sound, and do the data support the conclusions?

Reviewer #1: Partly

Reviewer #2: No

Reviewer #3: Yes

Reviewer #4: Partly

Reviewer #5: Yes

2. Has the statistical analysis been performed appropriately and rigorously?

Reviewer #1: No

Reviewer #2: No

Reviewer #3: Yes

Reviewer #4: Yes

Reviewer #5: I Don't Know

3. Have the authors made all data underlying the findings in their manuscript fully available?

Reviewer #1: Yes

Reviewer #2: Yes

Reviewer #3: No

Reviewer #4: No

Reviewer #5: Yes

4. Is the manuscript presented in an intelligible fashion and written in standard English?

Reviewer #1: Yes

Reviewer #2: Yes

Reviewer #3: Yes

Reviewer #4: No

Reviewer #5: Yes

Reviewer #1: The authors touch a very sensitive topic on the research innovations of PhD students in China. Using the data from PhD students in China the authors present a model that can be move inclusive, open ended and flexible for the well being and performance boosting of the graduate students.

How does this compares with scenarios in countries like USA or EU or India?

Is there a gender bias?

Given the changing guidelines and geopolitical influences, how do the authors see the change in the model in coming times?

A schematic of the overall work could been much more helpful in describing the story in simple figure format.

Reviewer #2: The authors have made a commendable effort, but there are several issues with the manuscript:

1. The manuscript requires further editing. For instance, line 34 contains the word "of," which renders the sentence unintelligible.

2. The introduction section lacks any references, despite numerous categorical statements made within it.

3. Demographic information should be presented in a tabular format for clarity.

4. Although the authors indicate they used recent graduates and those within two years of graduation, they should consider the stage of doctoral education when analyzing the results. This is important because students closer to graduation may experience increased pressure related to publication and post-graduation opportunities.

5. The discussion section needs to be rewritten for improved clarity and coherence. There is no attempt to discuss the results and relate it to similar studies.

6. There are several statistical analysis that can be explored by the authors from their data. They may want to consult a statistician.

Reviewer #3: Thank you for your valuable contribution. The manuscript is clear and well-written, and the methodology and findings are convincing. However, please briefly explain why soft inputs didn't directly impact dissertation quality. Also, mentioning any limitations regarding how the results might differ across various disciplines or types of universities, and clarifying the sampling criteria would strengthen the manuscript further. Finally, please make sure the data is fully publicly available as per the journal's guidelines.

Reviewer #4: The author has provided a strong theoretical contribution and challenges the linear input–output model commonly applied to higher education and the data is based on a large and representative sample of 1,500 doctoral students from 238 universities in China. However, the study is limited to Chinese doctoral students from selected "Double First-Class" universities which reduces the generalizability of the findings to other contexts, particularly to international or non-elite academic institutions. Further, as the study is focused on limited universities in China, there is limited cross-cultural comparison, which could provide more robust insights into how institutional strategies differ globally. The study focuses on institutional inputs only and gives limited attention to personal, social, or economic factors and the definition of academic performance is too narrow.

The manuscript has been associated with language errors also.

Reviewer #5: In the manuscript the Table 4 is mentioned (therefore there must be a table 1 and so on, ) but no tables shown. The manuscript is not showing the tables. Also on line 329 check the number format, seems it is quoting some other inputs related to the results.

**Do you want your identity to be public for this peer review?** For information about this choice, including consent withdrawal, please see our Privacy Policy

Reviewer #1: No

Reviewer #2: **Yes: ** Wasiu Balogun

Reviewer #3: No

Reviewer #4: **Yes: ** Anamika

Reviewer #5: **Yes: ** Magdalena Waleska Aldana Segura

---

## [Author Response · Author response to Decision Letter 1]

23 Jun 2025

Reviewer #1: The authors touch a very sensitive topic on the research innovations of PhD students in China. Using the data from PhD students in China the authors present a model that can be move inclusive, open ended and flexible for the well being and performance boosting of the graduate students.

How does this compares with scenarios in countries like USA or EU or India?

Is there a gender bias?

Given the changing guidelines and geopolitical influences, how do the authors see the change in the model in coming times?

A schematic of the overall work could been much more helpful in describing the story in simple figure format.

Response #1:

1.How does this compares with scenarios in countries like USA or EU or India?

This is a very constructive suggestion.In lines 44-55, we've added some relevant literature from American and European scholars.

2. Is there a gender bias?

In lines 202-209, we've added explanations that the 32.9% representation of women in the survey sample basically aligns with the average value 39.88% of PhD students in this sample during this period. In lines 223-225,we've also added ANOVA statistical analysis in gender. There are no significant inter-group differences.

3. Given the changing guidelines and geopolitical influences, how do the authors see the change in the model in coming times?

This is a good guide for future research. Honestly speaking, the cultivation of innovative talents is indeed affected by many factors, including policy guidelines and geopolitical influences and so on. In lines 548-549, we've cited it as a limitation in this research and tried to improve in the future.

4. A schematic of the overall work could been much more helpful in describing the story in simple figure format.

In order to better illustrate the overall work, we have added a data table(Table 1) on the overall survey.

Reviewer #2: The authors have made a commendable effort, but there are several issues with the manuscript:

1. The manuscript requires further editing. For instance, line 34 contains the word "of," which renders the sentence unintelligible.

2. The introduction section lacks any references, despite numerous categorical statements made within it.

3. Demographic information should be presented in a tabular format for clarity.

4. Although the authors indicate they used recent graduates and those within two years of graduation, they should consider the stage of doctoral education when analyzing the results. This is important because students closer to graduation may experience increased pressure related to publication and post-graduation opportunities.

5. The discussion section needs to be rewritten for improved clarity and coherence. There is no attempt to discuss the results and relate it to similar studies.

6. There are several statistical analysis that can be explored by the authors from their data. They may want to consult a statistician.

Response #2:

1. The manuscript requires further editing. For instance, line 34 contains the word "of," which renders the sentence unintelligible.

We are very sorry for this kind of error. We have corrected this error on line 32. We've also further checked the manuscript to avoid similar mistakes.

2. The introduction section lacks any references, despite numerous categorical statements made within it.

This is indeed a very constructive and wonderful comment for our research. In lines 35-55, we add a great deal of relevant references.

3. Demographic information should be presented in a tabular format for clarity.

We have added a data table(Table 1) with demographic information.

4. Although the authors indicate they used recent graduates and those within two years of graduation, they should consider the stage of doctoral education when analyzing the results. This is important because students closer to graduation may experience increased pressure related to publication and post-graduation opportunities.

This comment states exactly that students closer to graduation are more impornt in this research. As a matter of fact, the vast majority of the students were indeed at the graduating stage of doctoral education. We have added this explanation in lines 183-184.

5. The discussion section needs to be rewritten for improved clarity and coherence. There is no attempt to discuss the results and relate it to similar studies.

This is also a wonderful comment.We have rewrited the section of conclusion and discussion. We also succeeded in finding some representative similar studies in lines 453-457 and 473-483.

6. There are several statistical analysis that can be explored by the authors from their data. They may want to consult a statistician.

According to this recommendation,we've also consult statistician and added ANOVA statistical analysis in gender and discipline in lines 223-225.

Reviewer #3: Thank you for your valuable contribution. The manuscript is clear and well-written, and the methodology and findings are convincing. However, please briefly explain why soft inputs didn't directly impact dissertation quality. Also, mentioning any limitations regarding how the results might differ across various disciplines or types of universities, and clarifying the sampling criteria would strengthen the manuscript further. Finally, please make sure the data is fully publicly available as per the journal's guidelines.

Response #3:

1. Thank you for your valuable contribution. The manuscript is clear and well-written, and the methodology and findings are convincing. However, please briefly explain why soft inputs didn't directly impact dissertation quality.

We are very grateful to the reviewer for the compliments on our research.This motivates us to work harder to improve our research.To explain why soft inputs didn't directly impact dissertation quality, we've collected a large number of references and found organizational learning theory about two different logics of learning give a good eplanation in lines 452-463. The soft management of the university is more correlated with the executional learning or exploitative learning of doctoral students, which emphasizes the integration and application of existing resources. And soft inputs are the basis of a high-quality dissertation, but not to the decisive degree.

2. Also, mentioning any limitations regarding how the results might differ across various disciplines or types of universities, and clarifying the sampling criteria would strengthen the manuscript further.

We have clarified the gender criteria in lines 202-209 and the discipline criteria in lines 213-222. We've also made ANOVA statistical analysis in gender and discipline in lines 223-225.

3.Finally, please make sure the data is fully publicly available as per the journal's guidelines.

We have compiled the relevant data and attached them to the manuscript in supporting information.

Reviewer #4: The author has provided a strong theoretical contribution and challenges the linear input–output model commonly applied to higher education and the data is based on a large and representative sample of 1,500 doctoral students from 238 universities in China. However, the study is limited to Chinese doctoral students from selected "Double First-Class" universities which reduces the generalizability of the findings to other contexts, particularly to international or non-elite academic institutions. Further, as the study is focused on limited universities in China, there is limited cross-cultural comparison, which could provide more robust insights into how institutional strategies differ globally. The study focuses on institutional inputs only and gives limited attention to personal, social, or economic factors and the definition of academic performance is too narrow.

The manuscript has been associated with language errors also.

Response #4:

1. The author has provided a strong theoretical contribution and challenges the linear input–output model commonly applied to higher education and the data is based on a large and representative sample of 1,500 doctoral students from 238 universities in China.

This is an exciting comment. This makes us feel that our research has a certain positive meaning and encourages us to conduct more in-depth research. Thanks again.

2. However, the study is limited to Chinese doctoral students from selected "Double First-Class" universities which reduces the generalizability of the findings to other contexts, particularly to international or non-elite academic institutions.

We have already pointed out a total of 1,500 valid samples were obtained from three kinds of universities including world-class universities, universities featuring world-class disciplines, and other universities in lines 199-202. Therefore, our research is not only mainly from double first-class universities, but also from nearly a third of other ordinary universities. Therefore, the findings of the study are still quite generalizable.

3. Further, as the study is focused on limited universities in China, there is limited cross-cultural comparison, which could provide more robust insights into how institutional strategies differ globally. The study focuses on institutional inputs only and gives limited attention to personal, social, or economic factors and the definition of academic performance is too narrow.

This is a very accurate and constructive comment, which makes us realize that the shortcomings of our research are indeed limited by many factors such as geography, funding, etc., which really need to be improved. At the end of the manuscript, we express our awareness of the limitations of research and carry out follow-up research in future research using a broad framework and cross-cultural research.

4. The manuscript has been associated with language errors also.

For this problem, our research group and AJE have conducted further review of the manuscript to avoid errors as possible.

Reviewer #5: In the manuscript the Table 4 is mentioned (therefore there must be a table 1 and so on, ) but no tables shown. The manuscript is not showing the tables. Also on line 329 check the number format, seems it is quoting some other inputs related to the results.

Response #5:

1. In the manuscript the Table 4 is mentioned (therefore there must be a table 1 and so on, ) but no tables shown. The manuscript is not showing the tables.

We are very sorry that this is a problem, and we have relabeled the tables.

2. Also on line 329 check the number format, seems it is quoting some other inputs related to the results.

Thanks for the meticulous attention of reviewers.We've corrected this error on line 378.

Once again, We would like to express our sincere gratitude to the reviewers for their meticulous efforts.

---

## [Decision Letter · Decision Letter 1]

Hard Input, Soft Management and Student Engagement: How Institutional Actions Promote Innovation Ability And Academic Performance among Top Innovative Talent

PONE-D-25-08772R1

Dear Dr. Zhang,

We’re pleased to inform you that your manuscript has been judged scientifically suitable for publication and will be formally accepted for publication once it meets all outstanding technical requirements.

Kind regards,

Claudia Noemi González Brambila, Ph.D.

Academic Editor

PLOS ONE

Additional Editor Comments (optional):

Reviewers' comments:

Reviewer's Responses to Questions

**Comments to the Author**

Reviewer #1: All comments have been addressed

Reviewer #3: (No Response)

2. Is the manuscript technically sound, and do the data support the conclusions?

Reviewer #1: Yes

Reviewer #3: Yes

3. Has the statistical analysis been performed appropriately and rigorously?

Reviewer #1: Yes

Reviewer #3: Yes

4. Have the authors made all data underlying the findings in their manuscript fully available?

Reviewer #1: Yes

Reviewer #3: Yes

5. Is the manuscript presented in an intelligible fashion and written in standard English?

Reviewer #1: Yes

Reviewer #3: Yes

Reviewer #1: Accept the revised manuscript. The authors have addressed all the questions and the modified manuscript is now recommended for publication.

Reviewer #3: Thank you for your comprehensive revisions. The manuscript has improved significantly, and your explanations regarding soft vs. hard inputs and the statistical treatment of the data are now well-structured and presented. The integration of theoretical perspectives and the clarification of sampling details are commendable. The only minor note concerns the phrasing of data availability; although the data file is included, the statement might benefit from aligning more closely with journal expectations. Nevertheless, the current version is acceptable.

**Do you want your identity to be public for this peer review?** For information about this choice, including consent withdrawal, please see our Privacy Policy

Reviewer #1: No

Reviewer #3: No
